# Association between Ambient Air Pollution and MRI-Defined Brain Infarcts in Health Examinations in China

**DOI:** 10.3390/ijerph18084325

**Published:** 2021-04-19

**Authors:** Jing Wu, Yi Ning, Yongxiang Gao, Ruiqi Shan, Bo Wang, Jun Lv, Liming Li

**Affiliations:** 1Department of Epidemiology and Biostatistics, School of Public Health, Peking University Health Science Center, Beijing 100191, China; 15527305167@163.com (J.W.); ruiqi0119@bjmu.edu.cn (R.S.); lvjun@bjmu.edu.cn (J.L.); 2Meinian Public Health Institute, Peking University Health Science Center, Beijing 100191, China; bowang1981@163.com; 3Meinian Institute of Health, Beijing 100191, China; yongxiang.gao@meinianresearch.com

**Keywords:** air pollution, brain infarcts, China

## Abstract

The study aimed to evaluate the relationships between air pollutants and risk of magnetic resonance imaging (MRI)-defined brain infarcts (BI). We used data from routine health examinations of 1,400,503 participants aged ≥18 years who underwent brain MRI scans in 174 cities in 30 provinces in China in 2018. We assessed exposures to particulate matter (PM)_2.5_, PM_10_, nitrogen dioxide (NO_2_), and carbon monoxide (CO) from 2015 to 2017. MRI-defined BI was defined as lesions ≥3 mm in diameter. Air pollutants were associated with a higher risk of MRI-defined BI. The odds ratio (OR) (95% CI) for MRI-defined BI comparing the highest with the lowest tertiles of air pollutant concentrations was 2.00 (1.96–2.03) for PM_2.5_, 1.68 (1.65–1.71) for PM_10_, 1.58 (1.55–1.61) for NO_2_, and 1.57 (1.54–1.60) for CO. Each SD increase in air pollutants was associated with 16–42% increases in the risk of MRI-defined BI. The associations were stronger in the elderly subgroup. This is the largest survey to evaluate the association between air pollution and MRI-defined BI. Our findings indicate that ambient air pollution was significantly associated with a higher risk of MRI-defined BI.

## 1. Introduction

Globally, stroke was the second leading cause of death in 2017 [1], accounting for nearly 11% (6.2 million) of total deaths [2]. In China, stroke was the leading cause of death in recent years [3,4]. However, whereas symptomatic stroke is relatively easily recognized clinically, asymptomatic brain infarcts are often ignored [5]. Brain infarcts (BI) are commonly found in people who undergo magnetic resonance imaging (MRI) examinations and the prevalence of these infarcts was reported in 8–28% of participants in previous studies [6]. Most MRI-defined BI are asymptomatic and are not associated with stroke symptoms [7], but are considered a precursor of symptomatic stroke and may be associated with dementia [8,9,10]. Due to a lack of effective therapies for stroke and dementia, it is critical to evaluate risk factors to prevent or delay the occurrence of MRI-defined BI. Ambient air pollution is a major public health concern worldwide, especially in lowand middle-income countries [11]. According to the Global Burden of Diseases Study (GBD) 2019, 11.3% (2.92 million) of all women’s deaths and 12.2% (3.75 million) of all men’s deaths were attributable to air pollution [12]. Ambient air pollution also has various adverse health effects. The GBD 2013 showed that air pollution contributed 29.2% of stroke burden [11], and the literature reveals that air pollution was associated with brain health and central nervous system diseases [13,14]. Although previous studies linked genetic and lifestyle risk factors with stroke and MRI-defined BI [7,15], evidence regarding the associations of ambient air pollutants with MRI-defined BI is limited [16,17,18]. Previous observational studies were conducted in high-income countries with relatively good air quality but the results were inconsistent [16,17,18]. In addition, the sample size of these studies was relatively small (N < 2000) [16,17,18]. There are no studies on the association of air pollution with MRI-defined BI in China. Understanding the association of air pollution and MRI-defined BI is critical for prioritizing action to prevent stroke and dementia in China and globally, as well as for the management of air pollution. Therefore, in this study, we investigated the associations of air pollutants and MRIdefined BI using data from routine health examinations covering 174 cities in 30 provinces of China. 

## 2. Methods

### 2.1. Study Design and Study Population

For the present analysis, we used routine health examination data from participants who underwent brain MRI scans at Meinian Health Screening Center from 1 January–31 December 2018, the largest health screening organization covering nearly all provinces in the Chinese mainland. Each health screening center was equipped with well-trained professional staff and a unified standard examination protocol was established in all health screening centers. Most of the physical examinations in the health screening centers were group physical examinations, which are provided free of charge to employees as a type of company welfare. From 1 January to 31 December 2018, a total of 1,442,518 participants that visited the screening centers underwent a brain MRI. For those who attended more than two health examinations, only the most recent checkup data were included. After excluding participants who were aged <18 years old and those with missing values in age, sex, home address or air pollutants, a total of 1,400,503 participants (708,277 men and 692,226 women) aged ≥18 years from 310 health screening centers in 174 cities in 30 provinces of China were included in the analysis. Only anonymous information was used in the study. The study was approved by the Peking University Institutional Review Board with a waiver of informed consent (ID: IRB00001052-19077).

### 2.2. Exposure to Air Pollution

Data on concentrations of particulate matter (PM)_2.5_, PM_10_, nitrogen dioxide (NO_2_), and carbon monoxide (CO) were obtained from China’s National Urban Air Quality Realtime Publishing Platform during 2015–2017, periods just prior to the study. For each city, average annual concentrations of PM_2.5_, PM_10_, NO_2_, and CO were estimated by averaging the concentrations from 2015 to 2017. The monitoring data have been extensively used in China, and the details of data collection are described in previous studies [19,20].

### 2.3. Outcome Assessment

All participants underwent MRI examinations, adhering to a standardized scan protocol. We used axial T1 weighted images, T2 weighted images, proton density weighted images or fluid -attenuated inversion recovery sequences on 1 T or 1.5 T MRI scanners (more than 85% of MRI scanners are 1.5 T scanners). All MRI scans were assessed by two experienced radiologists and disagreements were solved by consensus through discussion. MRI-defined BI was defined as an area of abnormal signal intensity that lacked mass effect with a size ≥3 mm in diameter [7,21].

### 2.4. Covariates

Demographic characteristics, anthropometric parameters, blood samples, and medical histories for each participant were collected by trained health professionals during the health examinations. Data on the gross domestic product (GDP) per capita were collected from the China Statistical Yearbook for 2017. The daily mean air temperature and relative humidity in 2018 were derived from the China Meteorological Data Sharing Service System (http://data.cma.cn/) (accessed on 12 April 2021). The average smoking rate was extracted from a national survey of smoking prevalence published in Lancet Respiratory Medicine [22]. Hypertension was defined as systolic blood pressure ≥140 mm Hg, or diastolic pressure ≥90 mm Hg, or a history of hypertension. Diabetes was determined by a fasting level of plasma glucose ≥7.0 mmol/L or a history of diabetes. Dyslipidemia was defined as having any of the following: triglyceride level ≥2.3 mmol/L, total cholesterol level ≥6.2 mmol/L, high-density lipoprotein cholesterol level <1.0 mmol/L, low-density lipoprotein cholesterol level ≥4.1 mmol/L, or a history of dyslipidemia. Fat liver disease was detected by ultrasonography, performed by experienced technicians. Atrial fibrillation was identified by the examination of electrocardiograms. Renal disease was defined according to an estimated glomerular filtration rate of <60 mL/min/1.73 m^2^ [23].

## 3. Data Analysis

The characteristics of study participants according to MRI-defined BI were compared using analysis of variance (ANOVA) and the chi-squared test for continuous and categorical variables, respectively. We computed Spearman correlation coefficients for four air pollutants (PM_2.5_, PM_10_, NO_2_, and CO) and meteorological variables. Multivariate logistic regression models were used to estimate the associations between air pollutants and MRIdefined BI. When air pollutants were considered as continuous variables, we also report the odds ratios (ORs) and 95% CIs of MRI-defined BI for each 1 SD increase in the concentrations of air pollutants. We specified three incremental models that sequentially adjusted for potential risk factors of MRI-defined BI. First, we adjusted for age (continuous) and sex. We also adjusted for city-level gross domestic product (GDP). Next, we adjusted for temperature. The final model (model 4) was considered as the main model and further used in subgroup analyses and sensitivity analyses.

We conducted subgroup analyses to estimate whether the associations between air pollutants and MRI-defined BI varied by age (<65, ≥65 years old) and sex. We tested the interactions in the final model. To test the robustness of the results, we also conducted sensitivity analysis by further adjusting for the relative humidity, or further adjusting for the province-level average smoking rate, or further adjusting for the individual body mass index (<18.5, 18.5–23.9, 24.0–27.9, ≥28.0 kg/m^2^), hypertension, diabetes, dyslipidemia, atrial fibrillation, fatty liver disease, and renal disease.

In addition, we used restricted cubic splines to flexibly model the associations of air pollutants with MRI-defined BI. In the cubic spline models, we adjusted for age, sex, citylevel GDP, and temperature.

All analyses were performed using SAS 9.4 (SAS Institute, Cary, NC, USA) and R version 3.6 (http://www.r-project.org/) (accessed on 12 April 2021). Statistical significance was defined as two-sided *p* < 0.05.

## 4. Results

The characteristics of the study participants are shown in Table 1. A total of 1,400,503 participants were included in the study. The locations and city-level characteristics of the 174 cities are shown in Appendix A. Of the study participants, 95,671 participants (6.8%) were found to have brain infarcts in MRI scans. The mean age of study participants was 46.4 (SD 12.4) years and the mean age of participants with MRI-defined BI was 58.8 (SD 10.4) years. Compared with those without MRI-defined BI, participants with MRI-defined BI were more likely to be men, older, overweight or obese, and had higher prevalence of hypertension, diabetes, and dyslipidemia (all *p* < 0.001; Table 1).

The mean (SD) concentrations of air pollutants were 51.12 (15.61) μg/m^3^ for PM_2.5_, 88.93 (30.08) μg/m^3^ for PM_10_, 35.98 (10.07) μg/m^3^ for NO_2_, and 1.07 (0.29) mg/m^3^ for CO. These air pollutants generally had a moderate to high correlation with each other (Table 2). 

The associations between exposure to air pollutants and MRI-defined BI are shown in Table 3. The adjusted OR (95% CI) for MRI-defined BI was 2.00 (1.96–2.03) for PM_2.5_, 1.68 (1.65–1.71) for PM_10_, 1.58 (1.55–1.61) for NO_2_, and 1.57 (1.54–1.60) for CO, comparing the highest with the lowest tertiles of air pollutant concentrations. Each 1 SD increase in air pollutants was associated with increased risk of MRI-defined BI; the adjusted ORs (95% CIs) for PM_2.5_, PM_10_, NO_2_, and CO were 1.42 (1.40–1.43), 1.35 (1.34–1.36), 1.28 (1.27– 1.29), and 1.30 (1.29–1.31), respectively (Table 3).

Restricted cubic splines were used to visualize the associations between air pollutants and MRI-defined BI (Figure 1). Consistent with the findings of our primary analysis, we observed that the risk of MRI-defined BI increased with higher concentrations of PM_2.5_, PM_10_, NO_2_, and CO based on the cubic spline models.

In the exploratory subgroup analyses, the associations between air pollutants and MRI-defined BI were significant in all subgroups, though the associations were stronger in older age groups. This difference was also found in a stratified analysis by sex for the association between PM_2.5_ and CO and MRI-defined BI, with a stronger association observed in men (Table 4).

The results were generally robust in sensitivity analyses by further adjusting for the relative humidity, or further adjusting for the average smoking rate, or further adjusting for the body mass index, hypertension, diabetes, dyslipidemia, atrial fibrillation, fatty liver disease and renal disease (Table 5).

## 5. Discussion

In this study, we found that PM_2.5_, PM_10_, NO_2_, and CO were all associated with an increased risk of MRI-defined BI among 1.4 million health examinations in China. Among air pollutants, the strongest associations were observed for PM_2.5_. Furthermore, we observed that participants in older age groups exhibited a greater risk of MRI-defined BI as a result of exposure to air pollution. This is the largest study to examine the association between exposure to ambient air pollution and MRI-defined BI.

Previous reviews of observational studies showed that particulate and gaseous air pollutants were associated with elevated risk of stroke, cognitive impairment, dementia, and accelerated brain aging [13,24]. Several recent studies linked environmental epidemiology to MRI to evaluate the effects of air pollution on brain health [25,26]. Based on brain MRI findings, prior studies found that air pollutants were associated with a smaller total brain volume [16,18], reduced white matter volume [27,28], reduced deep gray mattervolume [16,28], and increased risks of brain atrophy [18,26]. All the studies implied that air pollutants may impact brain structure and function.

However, few studies have assessed the association of air pollution and MRI-defined BI. Published epidemiological studies were conducted in high-income countries and their conclusions remain controversial. Consistent with our findings, the Framingham Offspring Study found that a 2 μg/m^3^ increase in PM_2.5_ was associated with increased risk of covert brain infarcts (OR: 1.46; 95% CI: 1.10–1.94) among 934 participants aged ≥60 years and free of dementia and stroke [18]. However, a study conducted in four U.S. sites failed to observe significant associations between PM exposure (PM_10_ and PM_2.5_) and brain infarcts detected by MRI among 1753 participants with an average age of 76 years old [16]. Another study conducted in 1075 stroke-free participants aged ≥50 years from the Northern Manhattan Study also did not observe associations of ambient PM_2.5_ and NO_2_ with subclinical brain infarcts [17]. The sample size and ranges of air pollutants in the studies were relatively smaller; thus, the association between air pollution and MRI-defined BI might be difficult to detect. However, in this study, with larger geographic regions and a wider range of air pollution, we observed significant associations between air pollutants and MRI-defined BI. We also found the associations were stronger in elderly subgroups; this may be due to older adults, who are affected by a decline in physical functions, being susceptible to air pollution and various diseases.

The mechanisms underlying the associations between air pollution and MRI-defined BI remain unclear. However, some potential explanations can be provided. Firstly, exposure to air pollution may result in platelet activation and thrombosis [29,30,31]. Secondly, ambient air pollution may induce systemic inflammatory responses through increased release of plasma cytokines [32,33], which has been associated with increased risk of MRIdefined BI [34]. Thirdly, air pollution may trigger an elevation in blood pressure and increased insulin resistance, thereby synergistically heightening the risk of MRI-defined BI [35]. Finally, artery calcification [36] and endothelial dysfunction [37] caused by air pollution may contribute to brain MRI abnormalities, including MRI-defined BI [38,39]. Further studies are needed to confirm this finding and identify potential mechanism.

The study has notable strengths. The study included a large number of participants with MRI data from 174 cities in 30 provinces of China and a wide range of air pollutant concentrations, geographic regions, and socioeconomic status, which allowed us to explore the potential effect modification. This is the first epidemiological study to examine the association of air pollution with the risk of MRI-defined BI in China, which provides support for the association of air pollution and MRI-defined BI and strengthens the importance of reducing air pollution. Furthermore, using a large quantity of health examination data, we explored dose–response associations between particulate and gaseous air pollutants and the risk of MRI-defined BI.

Our study is not without limitations. The use of ambient air pollutants may have resulted in measurement error, although this non-differential misclassification is expected to underestimate the associations. Using health examination databases, we lacked information on individual education levels, income levels, and lifestyle and behavioral factors, which may have led to residual confounding. In addition, we did not collect the history of clinically defined stroke; thus, we did not distinguish symptomatic stroke and asymptomatic brain infarcts. However, most MRI-defined BI were covert BI, which is a precursor to overt stroke [7]. Including both covert and overt BI also better reflected the spectrum of MRI-defined BI in health examinations. Finally, given that our data from health examinations were for relatively healthy and younger participants, generalizations from the results should be made with caution.

## 6. Conclusions

In conclusion, we found that exposure to air pollutants (PM_2.5_, PM_10_, NO_2_, and CO) was associated with the risk of MRI-defined BI. 

## Figures and Tables

**Figure 1 ijerph-18-04325-f001:**
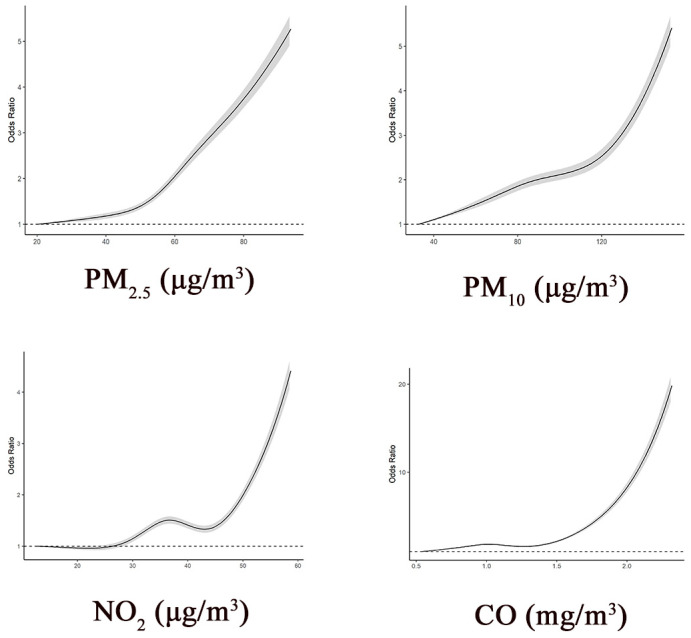
Associations between air pollutants and MRI-defined BI based on restricted cubic spline models. Note: Odd ratios are indicated by solid lines and 95% confidence intervals (CIs) by the gray area. The reference point is the lowest value for each air pollutant, with knots placed at the 5th, 25th, 50th, 75th, and 95th percentiles of each air pollutant. All models were adjusted for age (years), sex, city-level gross domestic product, and temperature. All *p*-values for non-linearity <0.001.

**Table 1 ijerph-18-04325-t001:** Characteristics of study participants with and without brain infarcts (BI) *.

Characteristics	Total	Participants without Brain Infarcts	Participants with Brain Infarcts
N	1,400,503	1,304,832	95,671
Age, year, mean ± SD	46.4 ± 12.4	45.5 ± 12.0	58.8 ± 10.4
Age group (years)			
<30	131,424 (9.4)	130,843 (10.0)	581 (0.6)
30–39	308,302 (22.0)	305,401 (23.4)	2901 (3.0)
40–49	373,217 (26.6)	361,220 (27.7)	11,997 (12.5)
50–59	372,543 (26.6)	338,641 (26.0)	33,902 (35.4)
60–69	171,891 (12.3)	138,997 (10.7)	32,894 (34.4)
≥70	43,126 (3.1)	29,730 (2.3)	13,396 (14.0)
Sex			
Female	692,226 (49.4)	650,169 (49.8)	42,057 (44.0)
Male	708,277 (50.6)	654,663 (50.2)	53,614 (56.0)
BMI (kg/m^2^)			
<18.5	36,842 (2.6)	35,675 (2.7)	1167 (1.2)
18.5–23.9	550,388 (39.3)	520,618 (39.9)	29,770 (31.1)
24.0–27.9	495,164 (35.4)	454,421 (34.8)	40,743 (42.6)
≥28.0	194,292 (13.9)	176,768 (13.5)	17,524 (18.3)
Hypertension			
No	963,201 (68.8)	924,303 (70.8)	38,898 (40.7)
Yes	368,445 (26.3)	315,745 (24.2)	52,700 (55.1)
Diabetes			
No	1,249,889 (89.2)	1,171,788 (89.8)	78,101 (81.6)
Yes	98,689 (7.0)	84,018 (6.4)	14,671 (15.3)
Dyslipidemia			
No	838,263 (59.9)	786,412 (60.3)	51,851 (54.2)
Yes	514,413 (36.7)	473,239 (36.3)	41,174 (43.0)
Atrial fibrillation			
No	1,315,385 (93.9)	1,225,927 (94.0)	89,458 (93.5)
Yes	2696 (0.2)	1995 (0.2)	701 (0.7)
Fatty liver disease			
No	809,925 (57.8)	762,058 (58.4)	47,867 (50.0)
Yes	533,952 (38.1)	489,412 (37.5)	44,540 (46.6)
Renal disease			
No	1,277,281 (91.2)	1,190,110 (91.2)	87,171 (91.1)
Yes	23,024 (1.6)	19,966 (1.5)	3058 (3.2)

Data were presented as *n* (%) or mean ± SD. *p* value was calculated by ANOVA (continuous variables) or χ^2^-test (categorical variables) for comparison of characteristics, and all *p* < 0.001. * There were 123,817 (8.8%), 68,857 (4.9%), 51,925 (3.7%), 47,827 (3.4%), 82,422 (5.9%), 56,626 (4.0%), and 100,198 (7.2%) missing values for body mass index (BMI), hypertension, diabetes, dyslipidemia, atrial fibrillation, fatty liver disease, and renal disease, respectively.

**Table 2 ijerph-18-04325-t002:** Spearman correlation coefficients among air pollutants, meteorological variables.

	Mean ± SD	Range	PM_2.5_ (μg/m^3^)	PM_10_ (μg/m^3^)	NO_2_ (μg/m^3^)	CO (mg/m^3^)	Temp (°C)	RH (%)
PM_2.5_ (μg/m^3^)	51.12 ± 15.61	20.20–93.70	1.00	0.92	0.74	0.68	−0.43	−0.49
PM_10_ (μg/m^3^)	88.93 ± 30.08	32.70–153.40	-	1.00	0.74	0.76	−0.59	−0.65
NO_2_ (μg/m^3^)	35.98 ± 10.07	12.60–58.70	-	-	1.00	0.59	−0.32	−0.50
CO (mg/m^3^)	1.07 ± 0.29	0.54–2.32	-	-	-	1.00	−0.52	−0.64
Temp (°C)	15.81 ± 4.44	2.96–24.66	-	-	-	-	1.00	0.74
RH (%)	69.10 ± 9.77	39.68–87.22	-	-	-	-	-	1.00

Abbreviations: CI, confidence interval; CO, carbon monoxide; NO_2_, nitrogen dioxide; PM, particulate matter; RH, relative humidity; Temp, temperature. All *p* < 0.01. Data source: China’s National Urban Air Quality Real Time Publishing Platform.

**Table 3 ijerph-18-04325-t003:** Odds ratio and 95% CI for the association between air pollution and brain infarcts.

	Case/N	Model 1 OR (95% CI)	Model 2 OR (95% CI)	Model 3 OR (95% CI)	Model 4 OR (95% CI)
PM_2.5_ (μg/m^3^)					
Tertile 1	20,150/467,596	1.00	1.00	1.00	1.00
Tertile 2	29,973/458,733	1.55 (1.52–1.58)	1.52 (1.49–1.55)	1.54 (1.52–1.57)	1.37 (1.35–1.40)
Tertile 3	45,548/474,174	2.36 (2.32–2.40)	2.24 (2.20–2.28)	2.20 (2.16–2.24)	2.00 (1.96–2.03)
*p*-trend		<0.001	<0.001	<0.001	<0.001
*Per SD increase*		1.50 (1.49–1.51)	1.48 (1.47–1.49)	1.46 (1.45–1.47)	1.42 (1.40–1.43)
PM_10_ (μg/m^3^)					
Tertile 1	21,474/473,040	1.00	1.00	1.00	1.00
Tertile 2	30,779/465,562	1.49 (1.46–1.52)	1.37 (1.34–1.40)	1.35 (1.32–1.38)	1.22 (1.20–1.25)
Tertile 3	43,418/461,901	2.18 (2.15–2.22)	2.02 (1.98–2.05)	1.96 (1.92–1.99)	1.68 (1.65–1.71)
*p*-trend		<0.001	<0.001	<0.001	<0.001
*Per SD increase*		1.45 (1.44–1.46)	1.42 (1.41–1.43)	1.41 (1.40–1.42)	1.35 (1.34–1.36)
NO_2_ (μg/m^3^)					
Tertile 1	24,936/477,987	1.00	1.00	1.00	1.00
Tertile 2	35,786/461,923	1.53 (1.50–1.55)	1.52 (1.50–1.55)	1.64 (1.61–1.67)	1.50 (1.48–1.53)
Tertile 3	34,949/460,593	1.49 (1.47–1.52)	1.52 (1.49–1.54)	1.73 (1.70–1.77)	1.58 (1.55–1.61)
*p*-trend		<0.001	<0.001	<0.001	<0.001
*Per SD increase*		1.24 (1.23–1.25)	1.26 (1.25–1.26)	1.35 (1.34–1.36)	1.28 (1.27–1.29)
CO (μg/m^3^)					
Tertile 1	21,278/466,523	1.00	1.00	1.00	1.00
Tertile 2	34,397/469,207	1.66 (1.63–1.69)	1.70 (1.67–1.73)	1.72 (1.69–1.76)	1.57 (1.54–1.60)
Tertile 3	39,996/464,773	1.97 (1.94–2.00)	1.90 (1.86–1.93)	1.84 (1.81–1.88)	1.57 (1.54–1.60)
*p*-trend		<0.001	<0.001	<0.001	<0.001
*Per SD increase*		1.37 (1.36–1.38)	1.37 (1.36–1.38)	1.35 (1.34–1.36)	1.30 (1.29–1.31)

Abbreviations: CI, confidence interval; CO, carbon monoxide; NO_2_, nitrogen dioxide; PM, particulate matter; SD, standard deviation. Model 1: Crude model. Model 2: Adjusted for age (years) and sex. Model 3: Model 2 plus city-level Gross Domestic Product. Model 4: Model 3 plus temperature. SD values are as follows: 15.61 μg/m^3^ for PM_2.5_, 30.08 μg/m^3^ for PM_10_, 10.07 μg/m^3^ for NO_2_, and 0.29 mg/m^3^ for CO.

**Table 4 ijerph-18-04325-t004:** Stratified analyses: Odds ratios and 95% CIs for the associations between each standard deviation increase in air pollutants and the risk of brain infarcts by age and sex.

	PM_2.5_	PM_10_	NO_2_	CO
Age, years				
<65	1.38 (1.37–1.39)	1.33 (1.31–1.34)	1.26 (1.25–1.27)	1.28 (1.27–1.29)
≥65	1.52 (1.49–1.54)	1.41 (1.39–1.44)	1.34 (1.32–1.37)	1.33 (1.31–1.35)
*p*-interaction	<0.001	<0.001	<0.001	<0.001
Sex				
Male	1.43 (1.41–1.44)	1.36 (1.35–1.38)	1.28 (1.26–1.29)	1.31 (1.30–1.32)
Female	1.40 (1.39–1.42)	1.34 (1.32–1.35)	1.29 (1.27–1.30)	1.28 (1.27–1.30)
*p*-interaction	0.02	0.13	0.13	0.03

Abbreviations: CI, confidence interval; CO, carbon monoxide; NO_2_, nitrogen dioxide; PM, particulate matter. All model adjusted for age (years), sex, city-level gross domestic product, and temperature. SD values are as follows: 15.61 μg/m^3^ for PM_2.5_, 30.08 μg/m^3^ for PM_10_, 10.07 μg/m^3^ for NO_2_, and 0.29 mg/m^3^ for CO.

**Table 5 ijerph-18-04325-t005:** Sensitivity analyses: odds ratio and 95% CI for the association between air pollution and brain infarcts.

	Model 1 OR (95% CI)	Model 2 OR (95% CI)	Model 3 OR (95% CI)
PM_2.5_ (μg/m^3^)			
Tertile 1	1.00	1.00	1.00
Tertile 2	1.42 (1.39–1.45)	1.39 (1.36–1.42)	1.45 (1.42–1.48)
Tertile 3	1.87 (1.83–1.91)	1.94 (1.90–1.98)	1.98 (1.94–2.02)
*p*-trend	<0.001	<0.001	<0.001
*Per SD increase*	1.39 (1.38–1.41)	1.40 (1.38–1.41)	1.40 (1.39–1.41)
PM_10_ (μg/m^3^)			
Tertile 1	1.00	1.00	1.00
Tertile 2	1.23 (1.20–1.25)	1.26 (1.23–1.28)	1.26 (1.24–1.29)
Tertile 3	1.49 (1.46–1.53)	1.66 (1.62–1.69)	1.73 (1.70–1.77)
*p*-trend	<0.001	<0.001	<0.001
*Per SD increase*	1.34 (1.33–1.36)	1.33 (1.32–1.34)	1.35 (1.34–1.36)
NO_2_ (μg/m^3^)			
Tertile 1	1.00	1.00	1.00
Tertile 2	1.39 (1.37–1.42)	1.46 (1.44–1.49)	1.58 (1.55–1.61)
Tertile 3	1.42 (1.39–1.45)	1.53 (1.50–1.56)	1.62 (1.59–1.66)
*p*-trend	<0.001	<0.001	<0.001
*Per SD increment*	1.22 (1.21–1.23)	1.26 (1.25–1.27)	1.29 (1.28–1.30)
CO (μg/m^3^)			
Tertile 1	1.00	1.00	1.00
Tertile 2	1.51 (1.48–1.54)	1.62 (1.59–1.66)	1.63 (1.60–1.66)
Tertile 3	1.29 (1.26–1.32)	1.57 (1.54–1.60)	1.61 (1.57–1.64)
*p*-trend	<0.001	<0.001	<0.001
*Per SD increase*	1.27 (1.26–1.28)	1.30 (1.29–1.31)	1.29 (1.28–1.30)

Abbreviations: CO, carbon monoxide; NO_2_, nitrogen dioxide; PM, particulate matter. Model 1 adjusted for age (years), sex, city-level gross domestic product, temperature, and relative humidity. Model 2 adjusted for age (years), sex, city-level gross domestic product, temperature and province-level average smoking rate. Model 3 adjusted for age (years), sex, city-level gross domestic product, temperature, and individual level variables, including body mass index, hypertension, diabetes, dyslipidemia, atrial fibrillation, fatty liver disease, and renal disease. Tests for a linear trend were calculated by fitting the median scores for quartiles as continuous variables in logistic regression models. SD values are as follows: 15.61 μg/m^3^ for PM_2.5_, 30.08 μg/m^3^ for PM_10_, 10.07 μg/m^3^ for NO_2_, and 0.29 mg/m^3^ for CO.

## Data Availability

All data used to support the findings of this study may be released upon application to the Meinian Institute of Health (Beijing, China), which can be contacted through Yi Ning (email: Yi.Ning@MeinianResearch.com).

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
