# Peer review of "Association between Ambient Air Pollution and MRI-Defined Brain Infarcts in Health Examinations in China"

_ijerph, 2021, doi:10.3390/ijerph18084325_

Round 1

Reviewer 1 Report

The study is interesting and well conducted.

It coud be an important reference for public health issues.

Please just implement the introduction citing other factors potentially influenced stroke and, in general elderly Patients: 

Biscetti, F.Giovannini, S.Straface, G., ...Landolfi, R.Flex, A.RANK/RANKL/OPG pathway: genetic association with history of ischemic stroke in Italian population
Giovannini, S.Onder, G.Leeuwenburgh, C., ...Bernabei, R.Landi, F.Myeloperoxidase levels and mortality in frail community-living elderly individuals
Giovannini, S.Onder, G.Lattanzio, F....Bernabei, R.Landi, F.
Selenium Concentrations and Mortality Among Community-Dwelling Older Adults: Results from ilSIRENTE Study

Author Response

Reviewer 1

Comments and Suggestions for Authors

The study is interesting and well conducted.

It coud be an important reference for public health issues.

Please just implement the introduction citing other factors potentially influenced stroke and, in general elderly Patients: 

Response: Thanks for your suggestion. We have added the information in the introduction section.
“Although previous studies have linked genetic and lifestyle risk factors with stroke and MRI-defined BI [7, 15], evidence regarding the associations of ambient air pollutants with MRI-defined BI is limited [16-18].”

Reviewer 2 Report

Ning et al. investigated the association between air pollution and MRI-defined brain infarction in subjects receiving health examinations. Several study suggestions should be acknowledged.

1. Please provide a map to illustrate the locations of 174 cities analyzed in the study area.

2. Atrial fibrillation, which is known as an essential risk factor for stroke, is not included in the baseline characteristics. The history of coronary artery disease, peripheral arterial disease, and transient ischemic attack is also not mentioned here.

3. Please provide a supplementary table to demonstrate the city-specific number of people, the number of residents, and the coverage rate based on 174 cities in this study.

4. Please provide a supplementary table to present the summary statistics on city-level characteristics in 174 cities, including the number of monitoring stations, annual average air pollution levels, annual average temperature, annual average relative humidity, and missing air pollution record, average age, and smoking rate.

5. As for the correlation between air pollutants, is Pearson correlation appropriate or Spearman correlation appropriate? Besides, the author should present the correlation between air pollutants and present the correlation between air pollutants, temperature, and relative humidity.

6. The temperature is adjusted in the regression model. However, the relative humidity is not controlled here. Several previous studies using the air quality monitoring process data usually present and adjust for relative humidity.

Author Response

Comments and Suggestions for Authors

Ning et al. investigated the association between air pollution and MRI-defined brain infarction in subjects receiving health examinations. Several study suggestions should be acknowledged.

  1. Please provide a map to illustrate the locations of 174 cities analyzed in the study area.

Response: Thanks for your suggestion. We have added a map to illustrate the locations of 174 cities in Supplementary Figure 1.

  1. Atrial fibrillation, which is known as an essential risk factor for stroke, is not included in the baseline characteristics. The history of coronary artery disease, peripheral arterial disease, and transient ischemic attack is also not mentioned here.

Response: Thanks for your suggestion.We did not collect the history of coronary artery disease, peripheral arterial disease and transient ischemic attack in the study. In the revised manuscript, the results were generally robust in sensitivity analyse of further adjusting for body mass index, hypertension, diabetes, dyslipidemia, atrial fibrillation, fatty liver disease, and renal disease (Table 5).

  1. Please provide a supplementary table to demonstrate the city-specific number of people, the number of residents, and the coverage rate based on 174 cities in this study.

Response: Thanks for your suggestion.We add a supplementary table to demonstrate city-level characteristics in 174 cities (Supplementary table 1).

  1. Please provide a supplementary table to present the summary statistics on city-level characteristics in 174 cities, including the number of monitoring stations, annual average air pollution levels, annual average temperature, annual average relative humidity, and missing air pollution record, average age, and smoking rate.

Response:Thanks for your suggestion.We add a supplementary table to demonstrate city-level characteristics in 174 cities (Supplementary table 1).

  1. As for the correlation between air pollutants, is Pearson correlation appropriate or Spearman correlation appropriate? Besides, the author should present the correlation between air pollutants and present the correlation between air pollutants, temperature, and relative humidity.

Response:Thanks for your suggestion.The Spearman correlation between air pollutants have been used in previous studies (Amini H, et al; Tian Y, et al.). We also add the information to present the correlation between air pollutants, temperature, and relative humidity (Table 2).

Amini H, Dehlendorff C, Lim YH, et al. Long-term exposure to air pollution and stroke incidence: A Danish Nurse cohort study. Environ Int. 2020 Sep;142:105891.

Tian Y, Liu H, Wu Y, et al. Association between ambient fine particulate pollution and hospital admissions for cause specific cardiovascular disease: time series study in 184 major Chinese cities. BMJ. 2019;367: l6572.

  1. The temperature is adjusted in the regression model. However, the relative humidity is not controlled here. Several previous studies using the air quality monitoring process data usually present and adjust for relative humidity.

Response:Thanks for your suggestion. In the revised manuscript, we furher adjusted for relative humidity, and the results were generally robust (Table 5).

Reviewer 3 Report

1 ..- in line 159, In table 1 a graph could be added that represents the information in a more digestible way for the reader.

2. The images in figure 1 are of poor quality and the values of the “x” and “y” axis are not appreciated.

3. The conclusions must mention the most outstanding results and the conclusions reached.

4. in line 30 the mentioned data must be updated since they mention investigations from 2017 to the past, currently there may be other causes of death.

5.- Of the pollutants in the air that are considered dangerous, a section must be added that indicates how they damage organisms, and what is the mechanism that these chemical compounds affect a cell or a group of cells.

Author Response

Comments and Suggestions for Authors

1 ..- in line 159, In table 1 a graph could be added that represents the information in a more digestible way for the reader.

Response: Thanks for your suggestion. In the revised manuscript, we added Supplementary table 1 and Supplementary Figure 1 to show more information for the reader.

  1. The images in figure 1 are of poor quality and the values of the “x” and “y” axis are not appreciated.

Response: Thanks for your suggestion. We added a high quality image in the revised manuscript.

  1. The conclusions must mention the most outstanding results and the conclusions reached.

Response: Thanks for your suggestion. We reviesed the conclusions section to mention the most outstanding results.

  1. in line 30 the mentioned data must be updated since they mention investigations from 2017 to the past, currently there may be other causes of death.

Response: Thanks for your suggestion. The information on concentrations of PM2.5, PM10, nitrogen dioxide (NO2), and carbon monoxide (CO) were obtained 2015-2017. In the study, the explosure were air pollutants and the outcome was MRI-defined brain infarcts. The information of MRI-defined brain infarcts were collected in 174 cities in 30 provinces of China in 2018.

5.- Of the pollutants in the air that are considered dangerous, a section must be added that indicates how they damage organisms, and what is the mechanism that these chemical compounds affect a cell or a group of cells.

Response: Thanks for your suggestion. We have added the mechanism in the Discussion section.

“The mechanisms underlying the associations of air pollution and MRI-defined BI remain unclear. However, some potential explanations can be provided. Firstly, exposure to air pollution may result in platelet activation and thrombosis [29-31]. Secondly, ambient air pollution might induce systemic inflammatory responses through increased release of plasma cytokines [32, 33], which has been associated with increased risk of MRI-defined BI [34].Thirdly, air pollution may trigger elevation in blood pressure and worsen insulin resistance, thereby synergistically heightening risk of MRI-defined BI [35]. Finally, artery calcification [36] and endothelial dysfunction [37] caused by air pollution may contributor to brain MRI abnormalities, including MRI-defined BI [38, 39]. Further basic medical studies are needed to provide insight into this casual pathway”

Round 2

Reviewer 2 Report

In the method of data analysis, Spearman correlation was mentioned. However, in the result section, Pearson correlation was mentioned. Which method was used in this study?

Author Response

In the method of data analysis, Spearman correlation was mentioned. However, in the result section, Pearson correlation was mentioned. Which method was used in this study?

Response:Thanks for your suggestion. I have used Spearman correlation coefficients in the paper and I have revised in the Table 1.The Spearman correlation between air pollutants have been used in previous studies (Amini H, et al; Tian Y, et al.).

Amini H, Dehlendorff C, Lim YH, et al. Long-term exposure to air pollution and stroke incidence: A Danish Nurse cohort study. Environ Int. 2020;142:105891.

Tian Y, Liu H, Zhao Z, et al. Association between ambient air pollution and daily hospital admissions for ischemic stroke: A nationwide time-series analysis. PLoS Med  2018, 15:e1002668.

Reviewer 3 Report

1 The conclusions were requested to be reviewed, and no significant change is seen, the conclusions do not have a relevant or complete content.
2.-Illustrate in a schematic way, what is the mechanism that these chemical compounds affect a cell or a group of cells, in the paper.

Author Response

1.The conclusions were requested to be reviewed, and no significant change is seen, the conclusions do not have a relevant or complete content.

Response: Thanks for your suggestion. We reviesed the conclusions section to mention the most outstanding results.

2.Illustrate in a schematic way, what is the mechanism that these chemical compounds affect a cell or a group of cells, in the paper.

Response: Thanks for your suggestion. Few studies have evaluated the associations of air pollution and brain infarcts detected by MRI. The mechanisms underlying the associations of air pollution and MRI-defined BI remain unclear. Our paper and previous studies [1-3] can only speculate about potential explanations. As a observation study, the mechanism is not the focus of the paper. Further studies are needed to confirm this finding and identify potential mechanism.

  1. Power MC, Lamichhane AP, Liao D, et al. The association of long-term exposure to particulate matter air pollution with brain MRI findings: the ARIC Study. Environ Health Perspect. 2018;126:027009.
  2. Kulick ER, Wellenius GA, Kaufman JD, et al. Long-term exposure to ambient air pollution and subclinical cerebrovascular disease in the Northern Manhattan Study. Stroke. 2017;48:1966-1968.
  3. Wilker EH, Preis SR, Beiser AS, et al. Long-term exposure to fine particulate matter, residential proximity to major roads and measures of brain structure. Stroke. 2015;46:1161-1166.